# *AnimeRun*: 2D Animation Visual Correspondence from Open Source 3D Movies

**Li Siyao**[1],    **Yuhang Li**[2,3],    **Bo Li**[1],    **Chao Dong**[2,4],    **Ziwei Liu**[1],    **Chen Change Loy**[1]✉

[1]S-Lab, Nanyang Technological University
[2]Shenzhen Institutes of Advanced Technology, Chinese Academy of Sciences
[3]University of Chinese Academy of Sciences    [4]Shanghai AI Laboratory
{siyao002, libo0013, ziwei.liu, ccloy}@ntu.edu.sg
{yh.li5, chao.dong}@siat.ac.cn

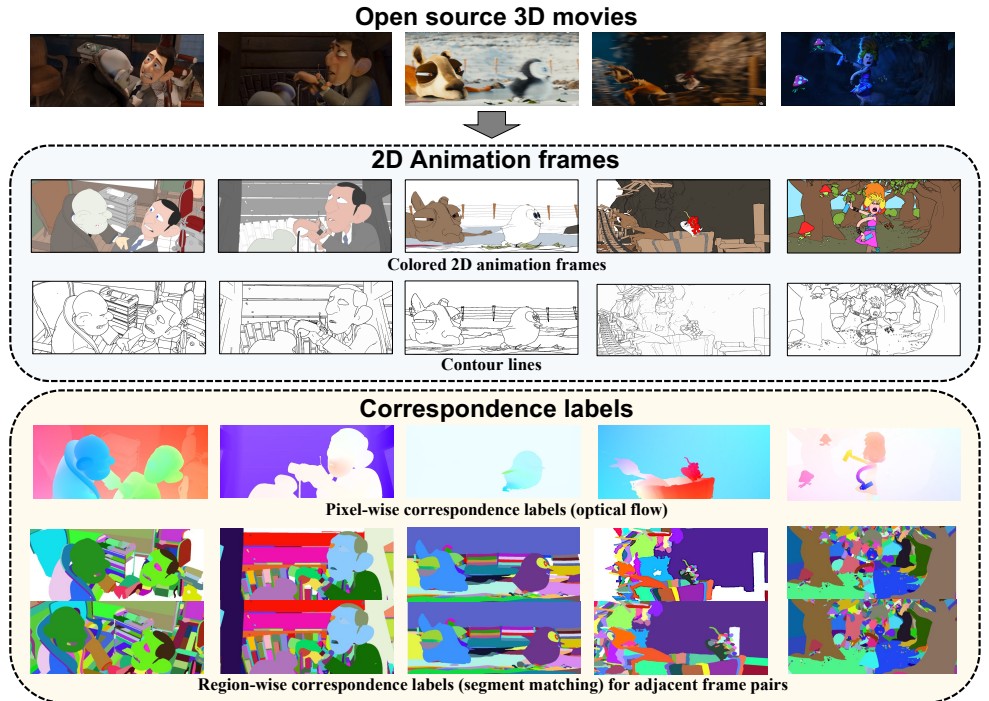

Figure 1: **AnimeRun.** We synthesize a rich correspondence dataset using open source 3D movies, with the input frames rendered in 2D cartoon style. We provide labels in both pixel-wise (optical flow) and region-wise (segment matching) levels for correspondence learning.

## Abstract

Existing correspondence datasets for two-dimensional (2D) cartoon suffer from simple frame composition and monotonic movements, making them insufficient to simulate real animations. In this work, we present a new 2D animation visual correspondence dataset, *AnimeRun*, by converting open source three-dimensional (3D) movies to full scenes in 2D style, including simultaneous moving background and interactions of multiple subjects. Our analyses show that the proposed dataset not only resembles real anime more in image composition, but also possesses richer and more complex motion patterns compared to existing datasets. With this dataset, we establish a comprehensive benchmark by evaluating several existing optical flow and segment matching methods, and analyze shortcomings of these methods on animation data. Data, code and other supplementary materials are available at https://lisiyao21.github.io/projects/AnimeRun.

✉Corresponding author

36th Conference on Neural Information Processing Systems (NeurIPS 2022) Track on Datasets and Benchmarks.

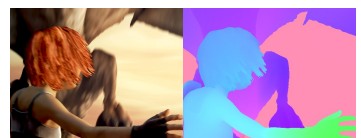
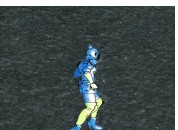
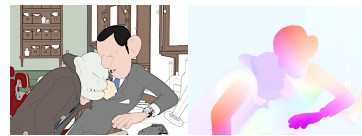

**(a) MPI-Sintel**
*Full scene* for 3D scenarios

**(b) CreativeFlow+**
*One model* for 2D cartoon

**(c) AnimeRun** (ours)
*Full scene* for 2D cartoon

Figure 2: **Image and optical flow in different datasets. (a) MPI-Sintel** [2] simulates the lighting and blur in natural scenarios within full scene composition. **(b)** Each frame sequence of **CreativeFlow+** [30] contains only one model rendered in cartoon-style with static background in different art styles. **(c) AnimeRun** (ours) features full scenes containing rich components and simultaneous moving background.

## 1 Introduction

Correspondence is the essence to many tasks in computer vision. For 2D animations, finding accurate visual correspondence of adjacent frames has a great potential for many real-world applications. For example, *region-wise* correspondence facilitates automatic colorization by propagating the color in a reference frame to corresponding segments in target frames [3], while *pixel-wise* correspondence (a.k.a optical flow) permits fine-grained image warping for frame interpolation [31, 4, 21]. Despite correspondence datasets for natural scenarios have been extensively explored [1, 7, 20, 2, 5, 19, 32], they are not well-suited for 2D cartoons due to the domain gap caused by inherently different imagery structure and properties. Specifically, natural images contain complex textures with diverse shading and blurring, while 2D cartoon is generally composited of flat color pieces with explicit contour lines.

To make in-domain correspondence data for animation, a viable way is to use computer graphics software (*e.g.*, Blender) to render 3D models into 2D style. In existing correspondence datasets designed for animation, *e.g.*, [30], a bunch of 3D models, including characters and objects, are collected from the Internet, and each time only a single model is rendered into a frame sequence with static background, with the model driven by its pre-defined motion. Since only one model needs to be rendered for a scene, this approach can synthesize large amount of data rapidly without specialized tuning for each scene. However, the "one-model" setting inevitably leads to monotonous composition and thus falls short in simulating real animations. In particular, there are a few shortcomings. **(1)** There are no simultaneous background or scene motions that are common in cartoon films. **(2)** Components (*e.g.*, segments) in a frame from only one character or object is much fewer than that from the full scene, unrealistically lowering the difficulty of the dataset compared to real animation. **(3)** There is no interactive movements (*e.g.*, fighting) between models as well as no occlusion between different objects, which are arguably the biggest challenge for correspondence learning in animation. **(4)** The pre-defined motions are usually monotonic and with a short temporal extent (*e.g.*, walking), which is not enough to simulate coherent actions in a long shot.

In view of these shortcomings, we propose a new dataset, *AnimeRun*, which to our knowledge is the first 2D animation visual correspondence dataset composed from *full scenes* of industry-level 3D movies. An overview of our dataset is shown in Figure 1. In *AnimeRun*, we provide both gray-scale contours and colored pictures for each frame to satisfy different usages on both the intermediate line arts and colorized animation. As to correspondence, we provide ground-truth labels in *pixel-wise* and *region-wise* levels, respectively, where the pixel-wise labels are dense optical flows of cartoon sequences, and the region-wise ones are the matching of segments in adjacent frames. In order to enrich the content, our dataset is built from three different open source films: *Agent 327* [26], *Caminandes 3: Llamigos* [27] and *Sprite Fright* [28], which feature not only a variety of changing scenes, including indoors, snow mountain, mine and forest, but also complex interactive patterns of motion. Transforming these full-scene data to 2D style is non-trivial as these movies come with complex and diverse 3D rendering effects, which are incompatible with the desired 2D cartoon style. To address this issue, on the premise of retaining the main characters and actions in the movie, we systematically perform a series of adjustments on *each film cut* to improve the compatibility with 2D animation styles and to ensure the accuracy of correspondence labels. Besides rendering the scenes and models based on the preset materials, we apply color augmentation following the statistics of real animation dataset [31] such that the synthetic data has a similar distribution of those in the wild. A detailed description to the design choice and label synthesis is demonstrated in Section 3.1 and 3.2.

Compared to existing datasets, *AnimeRun* has several advantages: 1) animation in full scene with simultaneous background motion, 2) richer image composition that is close to real animation, 3)

multiple objects/characters interaction and occlusion, 4) more complicated movement within larger magnitude, and 5) continuous motion in longer-story shots. A comparison on AnimeRun with previous datasets can be seen in Figure 2. The detailed statistics comparison appears in Section 3.3.

We split the AnimeRun dataset into training and test partitions. With the test set of *AnimeRun*, we prepare a series of benchmarks on the two kinds (region-wise and pixel-wise) of correspondence labels. To establish the benchmarks, we trained, and finetuned several existing methods on proposed training data as baselines. According to experimental results, we analyze the shortcomings of current methods on cartoon data from different fine-grained aspects. We also compare the effectiveness of CreativeFlow+ [30] and AnimeRun as a source for training. Experimental results show that our proposed data can benefit the prediction of accurate 2D animation correspondence for both the rich frame composition and the large motions compared to [30]. The detailed experimental settings and results are provided in Section 4.

In conclusion, the contributions of our work are three-fold: **(1)** we make the first full-scene animation correspondence dataset by converting open source 3D movies into 2D style. **(2)** We provide fine grained correspondence labels in both pixel-wise and region-wise levels. **(3)** We establish the first benchmark to correspondence in 2D animation.

## 2 Related Work

Optical flow has drawn much research attention in the last decade [1, 14, 9, 33, 38, 34, 12, 36, 42], while accurate flow prediction can burst many downstream tasks, *e.g.*, frame interpolation [16, 11, 24, 22, 37, 23, 15, 8]. As to optical flow data, since there is no sensor that can directly capture the actual flow values, the ground truth data of real scenes can only be made at a small scale [1] or be restricted on rigid objects [7, 20]. To cover more realistic motions, synthesizing real looking data using computer graphics tools becomes a feasible way. In Butler *et al.* [2], the authors proposed MPI-Sintel, an optical flow dataset rendered from an open source 3D animated movie, with dense ground-truth optical flows. To simulate real-world scenarios, Sintel is enhanced with natural features including complex lighting and shading, synthetic fog, and motion blur. In subsequent studies, a series of large-scale synthetic datasets, including FlyingChairs [5], FlyingThings3D [19], Virtual KITTI [6], VIPER [25] and REFRESH [18], are made for pretraining deep networks. More recently, Sun *et al.* proposed a new flow generation scheme that formulates the synthesis pipeline into a joint optimization problem, and constructed a new dataset, AutoFlow, to improve the pretraining process [32]. All these datasets aim for natural-looking representations in real scenarios rather than 2D cartoon.

For literature that considers styles in 2D animations, the most relevant work to ours is CreativeFlow+ [30], where a series of 3D models is rendered in diverse artistic styles like ink, pencil and cartoon. However, since each sample in CreativeFlow+ is composed of just one single model driven by simple predefined movements, it is inadequate to simulate real 2D animations. As to segment matching, Zhu *et al.* [43] annotated a small-scale dataset containing eight short sequences fewer than 150 frames in all, which is not enough to be a source of training. Zhang *et al.* [40] proposed large-scale segmentation labels for isolated cartoon images but without temporal correspondence. In this paper, we propose a dataset made from full-scene movies that possesses both rich components in image composition and complex motion patterns, with both pixel-wise and region-wise correspondence labels.

## 3 *AnimeRun* Dataset

One can collect correspondence data by using computer graphics software through synthesizing high-quality frame sequences and recording the associated ground-truth motions. However, since very few resources for 2D anime are made publicly available, converting 3D movies into 2D style becomes an appealing alternative to simulate 2D cartoons. In this section, we describe the design choices we made during the conversion process, including our definition to the 2D animation style and rendering settings (Section 3.1), and the mechanism of ground-truth correspondence label generation (Section 3.2). We further analyze the synthetic data by its imagery statistics and the correspondence labels. With that we compare our dataset with previous studies in Section 3.3.

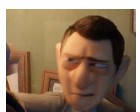 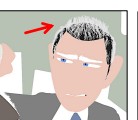 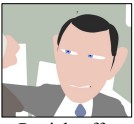 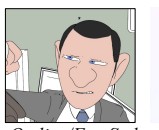 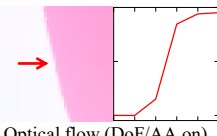 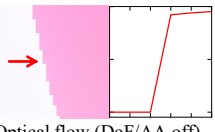

Original 3D    *FLAT* render    Particle off    *+ Outline/FreeStyle*    Optical flow (DoF/AA on)    Optical flow (DoF/AA off)

Figure 3: **Illustrations on some design choices.** In our data, 2D cartoon frames are rendered under the "*FLAT*" lighting of *Workbench* engine in Blender. Particle effects that simulate real small objects are turned off for a more similar looking of real cartoons. Depth of field (DoF) and Anti-Aliasing (AA) are turned off for sharp optical flow boundary.

## 3.1 Design Choices on 2D Animation Data

**Color, Line Art, and Texture.** A typical yet unique feature of 2D animation is that all subjects are outlined explicitly, and a closure of the contour lines is usually tinted into the same color. Following this observation, we use Blender's *Workbench engine* to render 3D models into flattened color segments, and synthesize the contour lines by enabling both *Outline* and *FreeStyle* options (see Figure 3). Since our dataset is aimed specifically for 2D cartoons instead of diverse styles, we do not add extra textures of artistic styles like watercolors or pencil drawings as in [30]. Besides, in order to obtain a similar color distribution to animations in the wild, we randomly assign representative colors of anime scenes in ATD-12K [31] to 3D models to augment the synthetic data (more discussion in the supplementary file). To provide the line art in gray scale separately, we render the movies by setting all materials to white, and apply a gamma correction with $\gamma = 0.2$ to enhance the contrast.

**Particle Effects and Environmental Assets.** When a 3D movie involves materials like hair, sweaters or sprays, particle effects can be exploited to simulate a mass of tiny and dense instances for a more realistic look. Under the 2D-style engine, such particle effects will be rendered into distinct solid lines, which do not fit the style of 2D animations (see Figure 3). To address this issue, we substitute the particles by instantiated objects to make synthetic data appear more similar to real cartoons. Apart from particle effects, invisible environmental assets, *e.g.*, fog block or emissive panels, are usually used to simulate naturalistic lights. Such assets are rendered as solid objects that occlude the main characters under the 2D-style engine, and therefore are removed in our data.

**Transparency.** Although transparency can appear in real 2D animation, as discussed in [2], translucent object may cause ambiguity for correspondence labels. Therefore, we follow previous optical flow dataset to remove all transparency cases and render all object as a solid instance.

**Bokeh Effects.** We find enabling depth of field (DoF) results in blur of flow values on the boundary between the object and the background as shown in Figure 3. Therefore, we exclude these two effects in our rendering settings.

**Camera Motion and Focal Length.** All three films contain a wealth of professional camera movements, including translation, rotation and subject tracking. However, when computing the motion vector, Blender will ignore the focal length change between adjacent frames and result in wrong optical flow. This problem is also reported as a bug in [2, 30]. To avoid inaccurate labels, we substitute such movements to positional shift along the direction with fixed focal length.

## 3.2 Ground-Truth Label Synthesis

**Optical Flow.** In addition to the settings mentioned above, we also turn off anti-aliasing by tuning the *pixel filter width* under *Film* option to the minimum value to obtain sharp motion boundaries, and turn off all illuminating objects like lamp and shining dots in eyes to avoid such objects disappearing in synthetic optical flow. After rendering, we perform quality check on each flow by using it to warp the target frame to the source frame to examine whether the frames are aligned. In general, the average pixel error between the source frame and the warped target frame is less than 0.01 when occluded regions are excluded.

**Occlusion Mask.** We compute additional occlusion mask from ground-truth optical flows following Shugrina *et al.* [30]. Specifically, given a pair of adjacent frames $I_s$ and $I_t$, with the forward flow $f_{s \rightarrow t}$ and the backward flow $f_{t \rightarrow s}$, for a point $\boldsymbol{x}$, if $\|f_{s \rightarrow t}(\boldsymbol{x}) + w_{f_{s \rightarrow t}}(f_{t \rightarrow s})(\boldsymbol{x})\| > 0.5$, where $w_{f_{s \rightarrow t}}$ denotes backward warping, then we mark $\boldsymbol{x}$ as occluded. The occlusion masks are further exploited for region-wise label generation and evaluations in optical flow benchmarks.

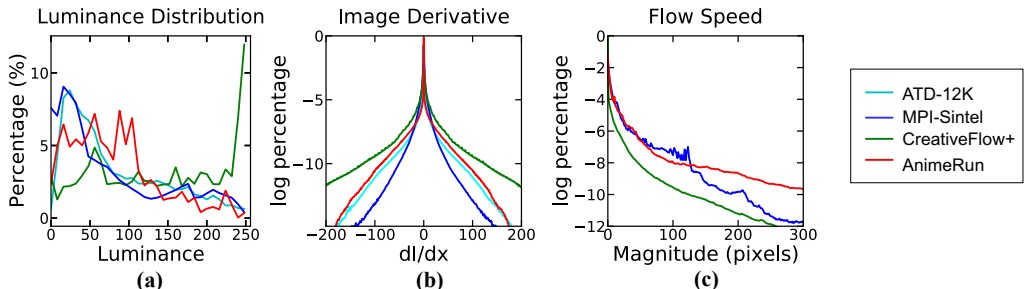

Figure 4: **Image and flow statistics. (a) Luminance histogram** of (color augmented) AnimeRun is closer to the real animes in ATD-12K [31] than CreativeFlow+ [30] (KL Divergence: 0.21 *vs.* 0.40 *vs.* 0.09). **(b)Log histogram of horizontal derivative** of AnimeRun is more similar to ATD-12K (summed absolute difference 253.42) compared to MPI-Sintel [2] (632.57) and CreativeFlow+ (1280.04). **(c) Log histogram on optical flow magnitude**.

**Cartoon Segmentation and Region-wise Matching Label.** To obtain region-wise correspondence labels, we first apply the trapped-ball filling algorithm [41] to divide all frames into segments along contour lines. Then, we derive the region-wise correspondence labels from the generated optical flows. Specifically, for segments $\{S_i^s\}$ and $\{S_j^t\}$ in the source and target frames, we mark $m_{s \to t}[i] = j$ if there exists a pixel $\boldsymbol{x} \in S_i^s$ and $\boldsymbol{x} + f_{s \to t}(\boldsymbol{x}) \in S_j^t$. Considering situations that an object is segmented into a single piece in the source frame, but becomes fragmented in the target frame due to occlusions, we record *all* segments $S_j^t$ that meet the above conditions and sort their priorities based on the percentage in the source region, where the one within highest priority is marked as the target for $S_i^s$. A segment $S_i^s$ will be marked as unmatched if no correspondence is found, and $m_{s \to t}[i]$ will be labeled as $-1$. To avoid mis-matching, when counting the *pixel-wise* correspondence in a segment, we mask off all pixels in the occlusion region, and erode the segment with a $3 \times 3$ kernel to reduce error caused by slight mis-alignment between segmentation edges and motion boundaries.

**Sequence Length and Image Resolution.** *AnimeRun* is composed of 2819 frames within 30 clips. Most clips consist of over 50 frames, where 9 of them contain more than 100 frames ($> 4$ sec). As to the composition of the three movies, *Agent 327*, *Caminandes*, and *Sprite Fright* contribute 741, 1565 and 513 frames, respectively. Except two clips in *Sprite Fright* that are rendered in 12 fps to prevent invalid unmoved frames, all film cuts are rendered in 24 fps. As to image resolution, we render at $960 \times 540$ for *Caminandes*, and at $1024 \times 436$ for *Agent 327* and *Sprite Fright*, which is in the same resolution of MPI-Sintel [2].

**Training and Test Split.** We provide a split for these clips to training set and test set, where the training set contains 1760 frames and the test set has 1059. For fair evaluation, we do not split continuous motion of one film cut into different subset to prevent algorithms simply multiplexing motion patterns in previous frames. In general, the two subsets share similar motion statistics, where the average motion magnitudes of the two subsets are both around 19 pixels.

**Data Structure and API.** We provide our data in a similar structure as MPI-Sintel. Each clip consists of a sequence of continuous frames $\{I^k\}$, contour lines $\{C^k\}$, segmentation label maps $\{S^k\}$, the forward and backward flows $\{f_{k \to k+1}, f_{k+1 \to k}\}$ and region-wise matching labels $\{m_{k \to k+1}, m_{k+1 \to k}\}$. A Python-based API is available at https://github.com/lisiyao21/AnimeRun.

### 3.3 Statistics and Comparison to Previous Datasets

**Image Statistics.** We evaluate the distributions of image luminance and derivative of AnimeRun and compare them to existing datasets. For luminance, pixel values ($[0, 255]$) are counted after converting colored images to gray scale. As shown in Figure 4(a), the histogram on real anime dataset ATD-12K [31] (cyan) reveals a bias towards dark colors, while CreativeFlow+ [30] (green) displays a larger proportion in high brightness, which yields a difference to the animation in the wild. In contrast, the color augmented data of AnimeRun has a closer distribution to ATD-12K, with a lower KL Divergence to ATD-12K than CreativeFlow+ (0.21 *vs.* 0.40). On image derivative, we compute the horizontal first-order difference and exhibit the log-histograms in Figure 4(b). Among compared datasets, MPI-Sintel involves motion blur and depth of field to simulate natural scenarios, which makes the image derivatives relatively lower than animations of ATD-12K; since a large part of

Table 1: **Region-wise statistics of different dataset**. *Since all flow values of background are zero, the average pixel shift of CreativeFlow+ is very small. Statistics for valid flows ($> 0.01$ pixel) are separately shown in brackets. $^{\dagger}$ The dataset for AnT is not publicly available, the number of segments is inferred to be less than 50 based on the description that "the maximum segment is 50" in [3]. Detailed statistics for each movie in AnimeRun are separately listed below.

| Dataset | # Seg. per Fr. | # Occ. Seg. per Fr. | Avg. Pix. shift per Fr. | Avg. Seg. shift per Fr. | Accum. Pix. shift per Seq. |
|---|---|---|---|---|---|
| ATD-12K [31] | **302** | – | – | – | – |
| CreativeFlow+ [30] | 49 | 0.82 | 3.84 (11.34)* | 16.40 | 373.68 (1106.01)* |
| AnT [3] (private) | $< 50^{\dagger}$ | – | – | – | – |
| AnimeRun (ours) | 237 | **9.59** | **19.27** | **19.09** | **1774.43** |
| *Agent 327* | 409 | 14.06 | 14.55 | 16.14 | 1478.94 |
| *Caminandes 3* | 152 | 8.17 | 24.53 | 22.39 | 2168.02 |
| *Sprite Fright* | 250 | 7.45 | 10.01 | 13.24 | 1118.86 |

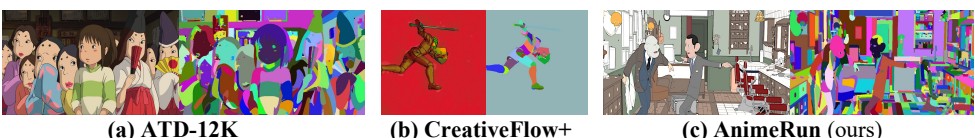

**(a) ATD-12K**            **(b) CreativeFlow+**            **(c) AnimeRun** (ours)

Figure 5: **Cartoon frames and segments of different datasets.** (a) ATD-12K [31] (real cartoon). (b) CreativeFlow+ [30]. (c) AnimeRun (ours).

frames in CreativeFlow+ [30] is rendered with high-frequency textures beyond 2D cartoons (*e.g.*, pencil sketches), it has more proportions of large gradient values in the distribution. Meanwhile, the log distribution of AnimeRun has a smaller summed absolute difference (SAD) to ATD-12K than Sintel and CreativeFlow+ (253.42 *vs.* 632.57 and 1280.04).

**Flow Statistics.** In the analysis of the pixel-wise correspondence label, we count the shift magnitude of each pixel on ground-truth optical flows, and compute the log-histograms as shown in Figure 4(c). As to a fair comparison among datasets in different resolutions, we interpolated all optical flows into $960 \times 540$ to unify the scale in the measurement. Generally, the flow magnitude of AnimeRun is larger than CreativeFlow+. While in comparison with Sintel, AnimeRun takes a slightly smaller proportion for movements less than 120 pixels (about 1/8 of the width or 1/4 of the height) and has more motions larger than this value.

**Region-wise Statistics.** Color segments are basic components of 2D animations, and a real cartoon frame can consist of hundreds of them as shown in Figure 5. Compared to existing datasets, AnimeRun composes of richer segments due to the rendering of full scenes. As shown in Table 1, the average number of segments in each frame reaches 237, which is closer to 302 in real animation data than that of other datasets. In contrast, the numbers of CreativeFlow+ and AnT do not exceed 50 due to only one subject appears in a frame sequence. Meanwhile, a frame in AnimeRun is expected to have 9.59 segments occluded in the next frame, which increases the richness of the correspondence labels and the complexity of the matching on it, while CreativeFlow+ has less than 1 on average for lacking interactive motions. Besides the rich quantity of segments, AnimeRun has larger shift distances in both pixel-wise and region-wise measurements. For pixel-wise shift, the average motion magnitude in AnimeRun is 19.27 pixels, which is 1.7 times that of CreativeFlow+. In the region-wise measurement, where the shift is computed as the distance between center points of matched regions, each segment is expected to move 19.09 pixels excluding the occluded ones, with 2.69 (16%) more than CreativeFlow+. In addition, we also calculate the accumulated pixel shift distance for each scene to evaluate motion magnitude in a film cut. Compared to predefined movements of a single subject in [30], The average motion of a clip in AnimeRun is more than 1.6 times that of CreativeFlow+ (1774.43 *vs.* 1106.01), which shows the advantage of movement in longer stories.

## 4 Benchmarks

We use AnimeRun to establish a benchmark on the *pixel-wise* and the *region-wise* correspondence. For pixel-wise correspondence, we perform evaluations using the two most landmark learning-based optical flow methods PWC-Net [33] and RAFT [34], and two recent methods GMA [12] and GMFlow

Table 2: **Quantitative results on pixel-wise correspondence.** occ./non-occ: occluded/non-occluded areas; line/flat: regions within/beyond 10 pixels to contour lines; $s$<10: ground truth flow speed < 10 pixels; $s$10-50: between 10 and 50 pixels; $s$>50: over 50 pixels. For each method, we separately test its original network weights and those finetuned on various datasets. T: FlyingThing3D [19]; S: Sintel [2]; Cr: CreativeFlow+ [30]; R: AnimeRun (ours).

| Method | EPE (pix.) ↓ | non-occ. | occ. | line | flat | $s$<10 | $s$10-50 | $s$>50 |
|---|---|---|---|---|---|---|---|---|
| PWC-Net [33] | 9.15 | 6.73 | 50.83 | 6.73 | 10.44 | 2.81 | 5.96 | 82.71 |
| PWC-Net ft. T+S | 7.36 | 5.15 | 45.46 | 5.51 | 8.36 | 1.51 | 4.61 | 74.19 |
| PWC-Net ft. T+Cr | 9.49 | 7.21 | 48.75 | 6.36 | 11.17 | 1.76 | 7.97 | 85.55 |
| PWC-Net ft. T+R | 7.04 | 4.96 | 42.02 | 5.14 | 8.07 | 1.24 | 4.64 | 71.40 |
| RAFT [34] | 5.40 | 3.68 | 35.05 | 3.75 | 6.29 | 0.80 | 3.19 | 58.25 |
| RAFT ft. T+S | 5.16 | 3.55 | 33.03 | 3.59 | 6.01 | 0.88 | 3.39 | 52.70 |
| RAFT ft. T+Cr | 6.66 | 4.64 | 41.39 | 4.29 | 7.93 | 0.90 | 4.28 | 70.51 |
| RAFT ft. T+R | 4.19 | 2.85 | 27.16 | 3.05 | 4.80 | 0.60 | 2.60 | 44.54 |
| GMA [12] | 5.63 | 3.83 | 36.81 | 3.99 | 6.52 | 0.85 | 3.11 | 61.92 |
| GMA ft. T+S | 5.27 | 3.60 | 34.22 | 3.72 | 6.11 | 0.84 | 2.96 | 57.25 |
| GMA ft. T+Cr | 6.27 | 4.32 | 39.98 | 3.96 | 7.51 | 0.91 | 4.22 | 64.84 |
| GMA ft. T+R | 3.86 | 2.61 | 25.38 | 2.92 | 4.36 | **0.58** | **2.43** | 40.61 |
| GMFlow [36] | 4.73 | 3.40 | 27.68 | 3.21 | 5.56 | 1.22 | 3.38 | 43.12 |
| GMFlow ft. T+S | 4.75 | 3.45 | 27.27 | 3.19 | 5.58 | 1.12 | 3.57 | 43.17 |
| GMFlow ft. T+Cr | 5.54 | 4.21 | 28.57 | 3.61 | 6.59 | 1.33 | 4.92 | 45.87 |
| GMFlow ft. T+R | **3.35** | **2.40** | **19.83** | **2.43** | **3.85** | 0.79 | 2.58 | **30.14** |

[36]. As to the region-wise one, we re-implement the most recent segment matching method AnT [3] as the baseline. For each benchmark, we compare the effectiveness of training on different datasets and discuss the shortcomings of existing methods as guidance for future improvement.

### 4.1 Pixel-wise Correspondence

**Baselines.** We conduct experiments on the two most representative deep learning based optical flow frameworks, *i.e.*, PWC-Net [33] and RAFT [34], which not only achieve state-of-the-art performance at the time they are published, but also trigger a lot of subsequent studies [17, 42, 39, 38, 10] to improve them as backbones. Besides, we evaluate GMA [12] and GMFlow [36] to show the merit of AnimeRun on the most recent algorithms.

For each method, in addition to evaluating the performance of the released pretrained weight, we perform finetuning using the training set of AnimeRun (denoted as "R") for 20 k iterations following the default settings of [34], and test the finetuned networks. Specifically, following the common practice in training optical flow networks, these networks are finetuned on a mixture of FlyingThings3D [19] (denoted as "T") and AnimeRun with a mixing ratio of about 1:10. The input data are cropped into $368 \times 768$ resolution and trained in a batch size of 12, with an initial learning rate of $10^{-4}$ and weight decay of $10^{-5}$. Besides, we conduct control experiments by finetuning the networks using CreativeFlow+ (denoted as "Cr") and Sintel (denoted as "S") within the same configuration[2] to verify whether the improvement of finetuning results from our dataset or from more training iterations, and compare the effectiveness of these datasets on animation.

**Evaluation Metrics.** We use the average end-point error (EPE) to measure the accuracy of predicted optical flow. For a more detailed analysis, we exploit the occlusion mask described in Section 3.2 to calculate the EPE of the occluded region (denoted as "occ.") and the non-occluded region (denoted as "non-occ.") separately. Since the large flattened regions in cartoons may cause matching uncertainty due to lack of texture, to see how algorithms perform on this kind of pattern, we count the EPE within (denoted as "line") and beyond (denoted as "flat") 10 pixels to contour lines, respectively. We also divide the measurement into three intervals ($s$<10, $s$10-50, $s$>50) according to the shift distance of the ground-truth optical flow to evaluate the performance on different flow magnitudes. The detailed quantitative results are shown in Table 2.

**Analysis. (1) GMFlow achieves the best overall performance.** As shown in Table 1, GMFlow achieves an average EPE of 3.35, improving 0.51 (13%) from the second place (GMA). Specifically,

---

[2]For CreativeFlow+ we set $480 \times 480$ as the image crop size.

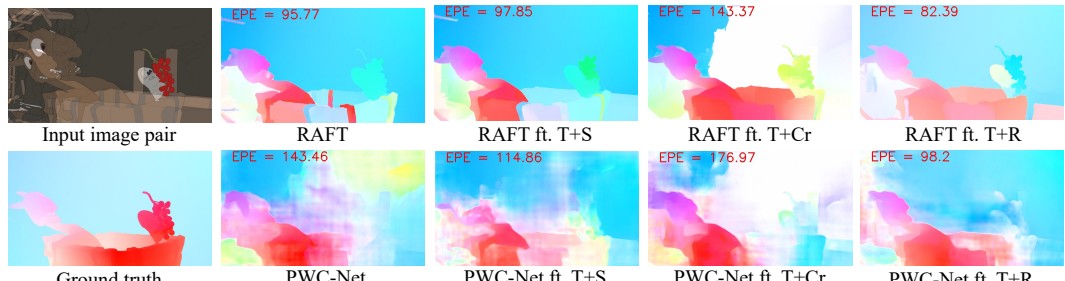

Figure 6: **Visualization of optical flow results.** Finetuning on AnimeRun (T+R) lowers the error of predicted optical flow on 2D animation compared to the original weights of networks and those finetuned on Sintel [2] (T+S) and CreativeFlow+ [30] (T+Cr).

the error of GMFlow on "$s$>50" is remarkably reduced by 10.47 (26%) than GMA, which could benefit from the global matching scheme [36]. While on the other hand, GMA behaves better on small motions. For motion shift smaller than 10 pixels, GMA achieves an EPE of 0.58, an improvement of 27% over GMFlow; as to "$s$10-50", GMA achieves the lowest error of 2.43. **(2) Finetuning on AnimeRun can improve the performance.** Compared to original network weights, overall performances of the four compared methods are improved by 2.11 (23%), 1.21 (22%), 1.77 (31%) and 1.38 (29%), respectively, after finetuned on AnimeRun; While referring to networks finetuned on Sintel ("ft. T+S"), the improvements are 0.32 (4%), 0.97 (19%), 1.41 (27%) and 1.40 (29%), respectively. As shown in Figure 6, finetuning on AnimeRun can fix some mismatching on the cart, where the original RAFT and PWC-Net predict incorrect flow values due to large distances. **(3) Finetuning on CreativeFlow+ lowers the performance on regions of large motions.** The performances of all four compared methods show a certain degree of decline after finetuned on CreativeFlow+. In particular, the overall EPE of "RAFT ft. T+Cr" gets 1.26 (23%) worse than the score of the original weight. If looking deeply into details, finetuning on CreativeFlow+ only slightly lower the performance on small shifts (0.10 for "s<10"), but significantly worsens that on large motion magnitudes (12.26 for "s>50"). Since CreativeFlow+ contains no simultaneous background motion, networks trained on it could tend to regress to zero when encountering a large flat area of the background. As shown in Figure 6, the flow values of the background region appear to be in the correct directions (visualized in blue) similar to ground truth for the original RAFT weights, but become zero (visualized in white) for "RAFT ft. T+Cr", yielding a poorer EPE value. **(4) The pixel-wise correspondence on the flat area is harder to predict than the area near contour lines.** As shown in the quantitative results, the average errors for "flat" regions are about 1.5 times larger than the corresponding EPEs for areas marked as "line", which applies in all compared methods. Different from natural images, it could be harder to find correspondence for regions beyond the line due to the lack of texture. Future work on animation optical flow can focus on improvement for such flat regions.

## 4.2 Region-wise Correspondence

**Baselines.** Instead of predicting the motion vector for each pixel, in region-wise correspondence, each segment is treated as a unit, and the goal is to find the corresponding segment in the target frame for each segment in the source frame. Therefore, the region-wise animation correspondence is a discrete matching problem similar to [29]. More recently, a deep learning-based framework, Animation Transformer (AnT) [3], is proposed to solve the segment matching problem for 2D animation. AnT is composited of a CNN-based segment descriptor, an MLP-based positional encoder, and a GNN-based feature aggregator realized in Transformer structure as SuperGlue [29], where the final mapping is computed from the segment-wise similarity matrix of aggregated features.

Since the source code of AnT is not publicly available, to set up the benchmark, we re-implement AnT and train it on the proposed AnimeRun and CreativeFlow+. Specifically, we use the Adam [13] optimizer to train the network for 3k iterations with a learning rate of $10^{-4}$ and a batch size of 16. Since each pair of input frames may consist of different numbers of segments, which is not capable of batch processing, we apply gradient accumulation for a batch. During training, input images are scaled and cropped to $368 \times 368$. For data augmentation, we perform random shifts and flip to input frames as well as their segmentation labels.

Table 3: **Quantitative results on AnimeRun region-wise correspondence benchmark.** The performances of AnT [3] trained on AnimeRun (R) is significantly higher than that trained on CreativeFlow+ (Cr).

| | Colored | | | | Contour Line | | | |
|---|---|---|---|---|---|---|---|---|
| Method | ACC (%) ↑ | non-occ. | occ. | #>300 | ACC (%) ↑ | non-occ. | occ. | #>300 |
| AnT tr. Cr | 70.64 | 72.88 | **0.00** | 58.71 | 68.29 | 70.41 | **0.00** | 51.93 |
| AnT tr. R | **80.61** | **83.27** | 0.00 | **74.79** | **80.84** | **83.46** | 0.00 | **75.80** |

Figure 7: **Input image pairs and visualization of segment matching.** The segments in the target frame (right of each image pair) are tinted into random colors, while the segments in source frame (left, red box) are tinted to the same color of matched segment in target, according to predicted matching labels. Occluded segments are colored in white. The source frames are expected to be colored as the same as that in ground truth.

**Evaluation Metrics.** We compute the average accuracy (ACC) of predicted correspondence. Besides, we count ACC values for occluded (where ground truth matching label is $-1$) and non-occluded segments, respectively. To measure the performance of image pairs within a large number of segments, we separately record the scores for cases that consist of more than 300 color pieces. The quantitative results are shown in Table 3.

**Analysis. (1) Training on AnimeRun significantly improves the matching accuracy compared to CreativeFlow+.** AnT trained on AnimeRun achieves an accuracy of 80.61% and 80.84% for segment matching on colored frames and contour lines, respectively, while the scores of that trained on CreativeFlow+ are about 10% lower. Specifically, for data containing more than 300 segments, "AnT tr. R" improves 16% and 20% than "AnT tr. Cr" for the two types of input, respectively, suggesting the effectiveness of training on AnimeRun that contains rich components.. **(2) Occlusion is difficult to recognize.** Although AnT achieves high overall scores, it performs poorly in recognizing a region that flew out of the scene or is occluded in the target frame. As shown in Table 3, AnT fails to pick out the occluded segments and obtains zero accuracies. **(3) Matching of repetitive or similar components is hard.** As shown in Figure 7, the background of the input image contains many similar longitudinally arranged wooden boards, where most of the predicted matching labels for these boards are incorrect due to their similar sizes and close positions. Although training on AnimeRun improves the accuracy on some noticeable parts (like hair), there are still many errors in such repetitive patterns. Therefore, future work can consider how to identify these similar components for more accurate matching.

## 5 Discussion

**Applications on Real Animation.** To explore the applicability of AnimeRun to real cartoons, we use optical flow networks finetuned on different datasets to conduct frame interpolation on ATD-12K [31]. Specifically, we substitute the motion prediction module of AnimeInterp [31] to RAFT [34] of various weights described in Section 4.1, and apply the subsequent warping and synthesis modules of the original AnimeInterp model to generate the intermediate frames. Following [31], besides evaluating the average PSNR and SSIM [35] between interpolated image and ground truth for each method, we report the scores of different difficulty levels defined in [31] separately in Table 4. As a baseline, the original RAFT achieves 29.20 dB of PSNR score on average, while that finetuned on Sintel [2] slightly improves 0.03dB. Finetuning on CreativeFlow+ [30] lowers the score by 0.01 dB; especially, it leads to a drop of 0.16 dB for the case marked as "difficult", which agrees with the poor performance of "$s$10-50" and "$s$>50" for "RAFT ft. T+Cr" in Table 2. In contrast, "RAFT ft. T+R" improves 0.15 dB on average, and improves 0.21dB, 0.16dB and 0.02dB for the three difficulty levels, respectively, which is consistent with the higher performance in Table 2.

Table 4: **Quantitative results of frame interpolation on ATD-12K.** We directly substitute motion prediction modules of AnimeInterp [31] to relative methods and evaluate the interpolation results. Difficulty levels are kept the same as [31]. T: FlyingThing3D [19]; S: Sintel [2]; Cr: CreativeFlow+ [30]; R: AnimeRun (ours).

| Method | Whole | | Easy | | Medium | | Hard | |
| --- | --- | --- | --- | --- | --- | --- | --- | --- |
| | PSNR (dB) ↑ | SSIM ↑ | PSNR | SSIM | PSNR | SSIM | PSNR | SSIM |
| RAFT [34] | 29.20 | 0.9558 | 31.35 | 0.9689 | 28.80 | 0.9566 | 26.65 | 0.9363 |
| RAFT ft. T+S | 29.23 | 0.9559 | 31.39 | 0.9691 | 28.83 | 0.9568 | 26.62 | 0.9361 |
| RAFT ft. T+Cr | 29.19 | 0.9561 | 31.41 | 0.9696 | 28.84 | 0.9571 | 26.48 | 0.9363 |
| RAFT ft. T+R | **29.35** | **0.9564** | **31.56** | **0.9698** | **28.96** | **0.9575** | **26.67** | **0.9364** |

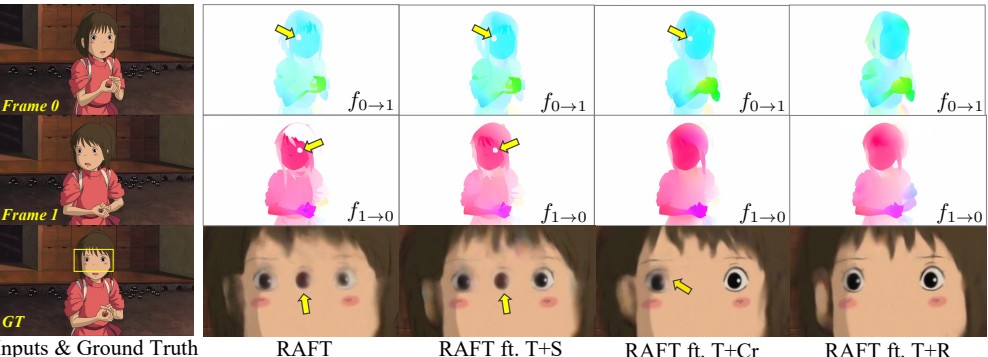

Figure 8: **Visualization of frame interpolation on ATD-12K [31]**. Given two input frames, the intermediate one is interpolated based on the forward ($f_{0\to1}$) and backward ($f_{1\to0}$) flows. For RAFT and "RAFT ft. T+S", flow errors in both $f_{0\to1}$ and $f_{1\to0}$ cause a wrongly interpolated eye; while the error in single $f_{0\to1}$ of "RAFT ft. T+Cr" yields blur of relevant part.

An example to illustrate the effectiveness of AnimeRun is shown in Figure 8, where *Chihiro* turns her head between two input frames. Since the original RAFT and that finetuned on Sintel [2] is trained on natural scene data, they fail to identify the difference between the left eye in frame 0 and the right eye in frame 1 under the textureless condition. That yields incorrect flow values at the eyes (indicated by arrows), which are not consistent with the rest facial parts, and further results in an additional eye in the interpolated frame. For "RAFT ft. T+Cr", although the values of $f_{1\to0}$ are consistent, the incorrectness in $f_{0\to1}$ leads to blurring of the left eye due to image warping errors. In contrast, "RAFT ft. T+R" can estimate the bidirectional flows more accurately and produce a clear interpolation result.

According to the above experiments, the scores in pixel-wise benchmark of AnimeRun are consistent to the quality of frame interpolation on ATD-12K [31], which implies AnimeRun can be a suitable optical flow quality indicator for animation in the wild. Meanwhile, as a source of training data, AnimeRun can improve the quality of interpolated results and therefore has great potential to facilitate future progress on relevant tasks.

**Summary.** We make the first full-scene 2D animation correspondence dataset consisting of continuous 2D cartoon frames and correspondence labels at pixel-wise and region-wise levels. The dataset is made by converting high-quality 3D movies to possess not only rich components in image composition but also larger and more complex motion, which can be a better simulation of real-world animations. Based on the dataset, benchmarks are established to analyze current solutions to correspondence problems for 2D animation. The proposed dataset not only has the potential to facilitate relevant cartoon applications as a source of training but also provides opportunities for future works for unsolved challenges analyzed in this paper.

**Acknowledgement.** This research is supported by the National Research Foundation, Singapore under its AI Singapore Programme (AISG Award No: AISG2-PhD-2022-01-031 (Siyao) and AISG2-PhD-2022-01-029 (Bo)). The study is also supported under the RIE2020 Industry Alignment Fund – Industry Collaboration Projects (IAF-ICP) Funding Initiative, as well as cash and in-kind contribution from the industry partner(s). This study is also supported by NTU NAP, MOE AcRF Tier 1 (2021-T1-001-088).

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
