# OpenReview forum: "AnimeRun: 2D Animation Visual Correspondence from Open Source 3D Movies"
_NeurIPS.cc/2022/Track/Datasets_and_Benchmarks — NeurIPS 2022 Datasets and Benchmarks _

### Official Review · Reviewer_BxAN · 2022-07-22
**Interesting animation dataset based on open-source animation**

**Rating:** 5
**Confidence:** 3

**Strengths:**

This article's clear strengths are the ability to use open-source animations and 3D to 2D processing workflows to obtain data for optical flow and segment matching tasks.

**Weaknesses:**

While this article is a valuable dataset introduction, there seem to be some key issues that should be discussed.

Mentioning a few practical aspects of the state-of-the-art methods and the datasets used might have been beneficial. Such information reduces the barrier to entry for experimentation.

There seems to be a lack of explicit Limitations section in this work, and parts of the writing seem incorrect.

**Additional Feedback:**

Some questions and issues worth discussing:
- When will the official API be available?
- Will the authors publish the pipeline for generating this dataset so that other researchers can reproduce and possibly use other 3D movies to generate 2D cartoon data.

**Clarity:**

This article is well written but contains many readability issues and unnecessary claims.

The following is a list of corrections that could be made:
- Abstract, lines 1; 3; 4 - when mentioning "2D", and "3D" it could be introduced as "two-dimensional (2D)", and "three-dimensional (3D)" upon first use.
- Abstract, lines 6-11 - instead of stating that "Statistics show", it would be more appropriate to state "Our analyses showed". This type of statement could directly introduce some metrics.
- Introduction, line 62 - there is a space before a comma.
- Introduction, line 62 - the sentence "To established the benchmark, we train or finetune", could be "To establish the benchmarks, we trained, and finetuned several existing methods on proposed training data as baselines.".
- Introduction, lines 65-72 - This content should be moved to two separate sections such as "Methods", or any comparable section, and "Conclusions". This text seems to fit there or be a part of abstract.
- Related work, line 75 - content "... or be restricted on rigid objects ...", should be "... or be restricted to rigid objects ...".
- Related work, line 80 - content "motion blurs", should be "motion blur".
- Related work, lines 82-83 - content "More recently, Sam et al. propose". Use past tense.
- Related work, line 84 - content "construct", use past tense "constructed".
- Related work, line 86 - content "For literature that considers styles", could be either "For literature that considered styling", or "For literature that considers styles".
- AnimeRun Dataset, lines 96-100 - introducing what each section contains might not be necessary, as each section contains that information within itself. This is, of course, a question of writing preference and convention.
- Section 3.1, lines 102-103 - starting the section with "Here, we present ...", might be unnecessary and equal to just presenting it.
- Section 3.1, line 116 - content "... realistic looking", could be "... realistic look".
- Section 3.1, line 120 - content "occludes", could be "occlude".
- Section 3.2, line 164 - content "On the sequence length ...", could start directly with "Most clips consist ...".
- Section 3.2, line 178 - content "A Python based API will be made available". If the API is not public yet, it is unnecessary to introduce it as something that may or may not happen in the future. On the other hand, if this is revised, the authors could consider mentioning some ongoing work on the official application programming interface (API).
- Section 3.3, line 179 - the title of the section should be changed to "Statistics and comparison to previous datasets" to reflect that more than one dataset is used in the comparison.
- Section 3.3, line 181 - content "to statistics of" could be deleted.
- Section 3.3, line 183 - content "reveals a bias to", should be "reveals a bias towards".
- Section 3.3, line 188 - content "motion blurs", should be "motion blur".
- Section 3.3, line 189 - remove "while".
- Section 3.3, lines 215-217 - this should be revised.
- Section 4.1, lines 226; 242; 251 - paragraph titles could be included without a full stop at the end.
- Section 4.1, line 229 - content "achieve", could be in the past tense.
- Section 4.1, line 253 - this should be revised.
- Section 4.2, lines 299; 305; 307; 310 - When using in-text enumeration, a full stop after each item is unnecessary and should be deleted.
- Section 5 - the conclusions are not clear. On the other hand, the authors listed conclusions at the end of the introduction.

The entirety of this article should be revised and restructured in some instances. The aim of this work could be clearly stated.

Instead of defining conclusions, authors can explore different article structures containing "Summary" or "Discussion" if applicable.

**Correctness:**

As the most valuable part of this publication is the data itself and the ability to use it for various modeling tasks, it seems that this publication is correct in its claims and does not exaggerate the value of the dataset.

**Documentation:**

It seems that the authors published a draft of the final project page. Unfortunately, the Python API is not published yet. Therefore, anyone that would like to use this dataset in its current state would have to write a custom dataset loading code.

The experiments and benchmarks performed in this article are also not published. The dataset is hosted on Google Drive, which is not a persistent scientific repository able to issue digital object identifiers (DOIs). There are multiple choices of such repositories, which include: Zenodo, Figshare, Dryad, and Open Science Framework, among others.

This work does not contain required documentation in the form of documentation mentioned on the "call for papers" page:
"Dataset documentation and intended uses. Recommended documentation frameworks include datasheets for datasets, dataset nutrition labels, data statements for NLP, and accountability frameworks."

**Ethics:**

This work deals with low-level visual analysis of 2D cartoons based on 3D movies. The authors used pre-existing public information. It is hard to imagine ethical considerations as it does not deal with sensitive data.

**Relation To Prior Work:**

The authors compare against other public and private datasets.

Moreover, benchmarks with existing methods were defined and showcased.

**Summary And Contributions:**

This submission introduces a new dataset and showcases its use for pixel-wise (optical flow) correspondence and region-wise (segment matching) correspondence tasks.

Authors achieve this by applying custom pre-processing on open-source 3D animation to transform it into 2D cartoons.

---

> ### Author Response · Authors · 2022-08-18
> **Response to Reviewer BxAN**
>
>
> **Q1: “ (This work) lack of explicit Limitations section”**
>
> **A1**: We have a “Limitations and Social Impacts” section in the original supplementary file. In the revised version, we elaborated more in detail.
>
>
> **Q2: “Comments and suggestions on writing to improve readability.”**
>
> **A2**: We appreciate the reviewer's suggestions to improve the readability of this work. We have revised the paper following the reviewer's comments. New/modified parts are colored into magenta.
>
>
> **Q3: “Code is not released yet.”**
>
> **A3**: We have uploaded the dataset API and the self-implemented AnT at https://github.com/lisiyao21/AnimeRun.
>
>
> **Q4: “Documentation is missing.”**
>
> **A4**: We add a datasheet for AnimeRun following the official direction of NeurIPS dataset track, which is included in the revised supplementary file.

---

> ### Author Response · Authors · 2022-08-29
> **Message to Reviewer BxAN**
>
> Dear reviewer,
>
> We appreciate your constructive comments and suggestions to improve our work. We have revised our paper according to your comments. Since it is the last day of the rebuttal period, would you please let us know whether we have resolved your concerns?

---

### Official Review · Reviewer_3aN8 · 2022-07-24
**The first practical dataset for 2D animation correspondence, room for improvement in manuscript**

**Rating:** 7
**Confidence:** 4

**Strengths:**

1. This paper tackles a less-explored but valuable task. 2D animation production is labor-intensive and requires a large number of manpower for the processes of frame interpolation and colorization. The dataset facilitates automation of those processes, leading to the well-being of workers and the preservation of 2D animation culture.

2. This paper introduces the first practical dataset for 2D animation correspondence. Existing datasets with real hand-drawn 2D animations (e.g., ATD-12K, [Zunda Horizon dataset](https://greenfunding.jp/pub/projects/1852), [TRIGGER dataset](https://www.nii.ac.jp/news/release/2022/0315.html)) do not have ground truth labels for correspondence. MPI-Sintel and CreativeFlow+ are not suitable for 2D animation (most animations draw full scene, not a single model like CreativeFlow+).

3. The effectiveness of the dataset is demonstrated from various perspectives. As shown in the statistics (Figure 4) and the quantitative results (Table 2, Table 3), the advantages of the proposed dataset over previous studies are clear. The analyses with visualization results in Sec. 4.1 and 4.2 are convincing.

**Weaknesses:**

1. Some parts of the manuscript are difficult to understand at a glance.
- Sec. 1: The real-world applications of the dataset are not so intuitive. They are described in text（L.13-17) without figures. It would be helpful to add a figure of the animation production process and explain that the two types of correspondence labels in the dataset are useful for frame interpolation and colorization. It would be good if the introduction is easy for anyone to understand (e.g., optical flow researchers not familiar with 2D animation, people involved in animation production who are not familiar with computer vision).
- Figure 2: Readers should be able to understand the advantages of this study at a glance without having to read the caption or main text. For example, with red and green textcolor,
```
  (a) MPI-Sintel      (b) CreativeFlow+  (c) AnimeRun (ours)
  Full scene for 3D   One model for 2D   Full scene for 2D
```
- Table 2 etc.: Dataset names are too abbreviated. I think it fits in the table without abbreviating anything other than FlyingThings3D. The comparison of Sintel, CreativeFlow+, and AnimeRun, which is important for this paper, should be understood by readers at a glance.

2. There is room for improvement in dataset creation.
- There is a large gap with real hand-drawn 2D animations (e.g., Figure 3 of ATD-12K [19]). Specifically, AnimeRun lacks shade and background texture. Animations and games with toon-shaded 3D models look more like real hand-drawn 2D animations (e.g., [D4DJ](https://www.youtube.com/watch?v=u5_dOnl-UJE), [Expelled from Paradise](https://www.youtube.com/watch?v=89heoIzvtUw), [Umamusume](https://www.youtube.com/watch?v=cmuTy73jzSs), [Genshin Impact](https://www.youtube.com/watch?v=PXp8qKZeHeY)).
- The dataset is made from only three videos, and all three were produced by Blender Studio.
- L.104-109: AnimeRun neglects the diverse styles of 2D cartoons and focuses on typical styles. Unlike CreativeFlow+, it lacks diversity in line drawing styles. Although the stylistic diversity of CreativeFlow+ looks excessive, it would be possible for the authors to make an effort to get closer to the distribution of real animation styles.
- If the dataset only aims for applications to midway processes of animation production (e.g., frame interpolation and colorization) and there is no need to resemble finished products, the authors should state this clearly. In that case, "finished animation" (L.40) should be modified.
- If the authors do not address the above limitations, they could be described in Supplementary File.

**Additional Feedback:**

- Can the authors provide depth and normal as in CreativeFlow+? If they are provided in another zip file, other researchers may use them in future research.
- Figure 4(a) The result of MPI-Sintel or the reason for omission should be shown.
- Figure 4(a) The curve of AnimeRun is not as smooth as that of ATD-12K. Is there any reason? If the authors cannot resolve the dataset bias, they can recommend luminance augmentation to dataset users.

Typos and other writing issues:
- The authors could check that word usage is consistent within the paper (specifically, "subjects", "models", "characters", and "objects").
- With or without hyphen for "full scene", "fine grained", "open source"
- "benchmarks" or "a benchmark"
- L.29 etc.: Is "." of "(1)." needed?
- L.42: "region-wise and pixel-wise levels, respectively" -> "pixel-wise and region-wise levels"
- L.62: "labels ." -> "labels."
- L.86: "consider" -> "considers"
- L.87: "a series of 3D models are rendered" -> "a series of 3D models is rendered" or "3D models are rendered"
- Figure 3: What does "FLAT" mean?
- Figure 3: "Particle effect" -> "Particle effects"
- L.104 etc.: "Color, Line Art and Texture." -> "Color, Line Art, and Texture." Oxford comma
- L.116: "Under 2D-style engine" -> "Under the 2D-style engine" if the issue is specific to the Blender's Workbench engine.
- L.146: "Shuegrina" -> "Shugrina"
- L.146: It will be kind to add reference to [the code](https://github.com/creativefloworg/creativeflow/blob/master/creativeflow/blender/compute_occlusions_main.py) or [Errata and Data Details](https://www.cs.toronto.edu/creativeflow/files/2596-errata.pdf).
- L.159: What "m" denotes should be written around this line.
- L.159: "-1" -> "``$-1$''"
- L.170: "Training and Test Split" -> "Training and Test Split."
- L.173: "two subset" -> "two subsets"
- L.173: "statistics ,where" -> "statistics, where"
- L.178: "Python based API" -> "Python-based API"
- L.179: "Dataset" -> "Datasets"
- L.182 etc.: "grayscale" and "gray scale" should be consistent.
- L.190: "are rendered" -> "is rendered"
- L.197-200: Long sentense
- Table 1 and Figure 5: Too close
- Figure 5: "ATD-12k" -> "ATD-12K"
- Table 2 caption: "s" -> "$s$"
- L.220: "respectively" is unnecessary.
- L.226: "Baseline Method settings." -> "Baseline method settings."
- L.227: "optimal" -> "the best"
- L.234: "20K" -> "20\\,k"
- L.252: "According to the experimental results," is unnecessary.
- L.254: "fintuned" -> "finetuned"
- L.258: "(3)" -> "(2)"
- L.288: "respectively" is unnecessary.
- L.288: "the Adam optimizer" lacks citation.
- L.289: "3K" -> "3\\,k"
- Table 3: "Quantitative Results on AnimeRun region-wise Correspondence Benchmark." -> "Quantitative results on AnimeRun region-wise correspondence benchmark."
- Table 3 and Figure 7: Too close
- L.295: "-1" -> "``$-1$''"
- L.296: "respectively" is unnecessary.
- L.303: "`AnT tr. R''" -> "``AnT tr. R''"
- L.305: "According to experimental results," is unnecessary.
- L.309: "or is occluded or in the target frame" -> "or is occluded in the target frame"
- References: Please check middle names (e.g., "Michael J Black" -> "Michael J. Black")
- "KITTI" -> "{KITTI}", "CNNs" -> "{CNNs}", etc.
- @inproceedings vs. @article
- [2]: "ECCV. Springer," -> "ECCV," (for consistent style)
- [6] and [7] could be unified to [7] (see the description of "Citation" in http://www.cvlibs.net/datasets/kitti/ )
- [23]: "NeurIPS, 32," -> "NeurIPS,"
- Checklist 3(d): Where?
- Checklist 4(b): Where?
- Supplementary File L.13: Words other than "general" may be appropriate (e.g., typical, representative, standard).
- Supplementary File L.16-17: Even datasets made from public data can fuel ethical issues.

**Clarity:**

Moderate. I hope the authors will improve the manuscript during the rebuttal period.

**Correctness:**

The claims and evaluation seem correct. However, there is no way to verify correctness and reproducibility. If the authors publish codes for dataset construction and experiments, other researchers will be able to verify correctness. In particular, the correctness of the re-implementation of AnT (L.287) should be verified by other researchers.

**Documentation:**

Documentation for the dataset is insufficient. The authors should include intended uses, licenses, and maintenance plan, possibly in Supplementary File.

**Ethics:**

Since the dataset is synthetic data converted from fiction movies, it has less concerns than real-world data. However, synthetic data could contain offensive contents. The authors should discuss this specifically in Supplementary File and modify Checklist 4(e). For example,
- None of the movies in the dataset are age-restricted on YouTube.
- Considering the nature of the optical flow task and the industrial application of the dataset, it is reasonable to include characters interaction in battle scenes.
- We excluded excessively violent scenes.

**Relation To Prior Work:**

Difference from the most relevant work (CreativeFlow+ and MPI-Sintel) is clear, although there is room for improvement in figures and tables.

The authors could refer to the following datasets as a prior or complementary dataset and describe the difference explicitly in Sec. 2. Haichao Zhu et al. [a] manually annotated the region-wise correspondence labels of eight animation sequences. Lvmin Zhang et al. [b] claims DanbooRegion benefits cartoon tracking like [a], although they neither provide spatiotemporal labels nor experimental results.

If the authors cite [c] in addition to [19] in L.17, readers can recognize that contour lines are useful for line drawing frame interpolation.

[a] Haichao Zhu et al., Globally Optimal Toon Tracking, TOG, 2016. http://www.cse.cuhk.edu.hk/~ttwong/papers/toontrack/toontrack.html
[b] Lvmin Zhang et al., DanbooRegion: An Illustration Region Dataset, ECCV, 2020. https://www.ecva.net/papers/eccv_2020/papers_ECCV/html/1816_ECCV_2020_paper.php   https://lllyasviel.github.io/DanbooRegion/
[c] Rei Narita et al., Optical Flow Based Line Drawing Frame Interpolation Using Distance Transform to Support Inbetweenings, ICIP, 2019. https://ieeexplore.ieee.org/document/8803506

**Summary And Contributions:**

This paper introduces AnimeRun, the first practical dataset for 2D animation correspondence. Unlike existing datasets, it is suitable for full-scene 2D animation. It contains pixel-wise correspondence labels for optical flow estimation and region-wise correspondence labels for segment matching. Using the dataset, the authors establish benchmarks for the two types of correspondence. The results show the effectiveness of AnimeRun and the limitations of existing methods.

---

> ### Author Response · Authors · 2022-08-18
> **Response to Reviewer 3aN8**
>
>
> **Q1: “Comments and suggestions to improve the readability.”**
>
> **A1**:  We appreciate the reviewer’s  suggestions to improve the readability of this work. We have revised the paper following the reviewer's comments including:
> - The animation production process is illustrated and explained in revised supplementary file (Sec 2, Fig 2);
> - Highlight features of AnimeRun in Fig 2 more explicitly;
> - Definitions of abbreviations in captions of Tab 2,3,4.
>
> New/modified parts are colored in magenta.
>
>
> **Q2: “There is room for improvement in dataset creation (for special effects and diverse styles).”**
>
> **A2**:
> Indeed, AnimeRun does not include diverse special effects. As discussed in the revised supplementary file (L. 26-33, Fig. 2), AnimeRun is designed to capture a generic style of 2D cartoons without composition and special effects such as bokeh, shading or blurring. Such properties make AnimeRun well-suited for intermediate stages in the anime production line that involves contour lines and colored frames. It is noteworthy that while lacking such  special effects, AnimeRun is still beneficial for finished animation. In the Discussion part of the revised paper, we show that AnimeRun offers more accurate frame interpolation on real cartoon data than baselines.
>
> **Q3: “Code is not released, especially that for AnT”**
>
> **A3**: We have uploaded the API and self-implemented AnT. Please see https://github.com/lisiyao21/AnimeRun.
>
> **Q4: “The authors could refer to the following datasets as a prior or complementary dataset…”**
>
> **A4**: We have referred to these works in Sec 2 following the reviewer’s suggestions, and discussed differences between these datasets and AnimeRun.
>
> **Q5: “Documentation for the dataset is insufficient.”**
>
> **A5**: We added a datasheet for AnimeRun following the official direction of NeurIPS dataset track, which is included in the revised supplementary file. In Particular, we illustrate that our data don’t contain offensive content in L.71-77 of the datasheet in revised supplementary files.
>
> **Q6: “More labels (depth, normal) of animation data.”**
>
> **A5**: We did make depth labels. We will release it after checking its correctness.

---

> > ### Comment · Reviewer_3aN8 · 2022-08-20
> > **Response to authors**
> >
> > Thank you for the great effort for the responses and revision.
> > My concerns have been largely addressed.
> > I have raised my rating.
> >
> > Some writing issues and minor comments:
> > - Figure 2: I expected One model (undesirable property) to be written in red and Full scene (desirable property) to be written in green.
> > - L.248-268 "(1).", "(2).", "(3).", "(4).": "." is unnecessary.
> > - Table 4: "Quantitative Results of Frame Interpolation on ATD-12K" -> "Quantitative results of frame interpolation on ATD-12K"
> > - Figure 8: "Visualization of Frame Interpolation on ATD-12K" -> "Visualization of frame interpolation on ATD-12K"
> > - "&Ground" -> "& Ground"
> > - "froward" -> "forward"
> > - L.322 and L.325: "0.03dB", "0.21dB, 0.16dB and 0.02dB" lack space before dB.
> > - References [6]: It will be appropriate to cite the KITTI paper of CVPR 2012.
> > - Supplementary File L.67: "as shown in 2." -> "as shown in Figure 2."
> > - Datasheet L.115 and L.129: Which version of CC BY-NC License should be clarified.

---

> > > ### Author Response · Authors · 2022-08-26
> > > **More response to Reviewer 3aN8**
> > >
> > > We thank the reviewer's recognition on our work and comments to improve the writing! We have further polished the writing following the review and revised the paper and supplementary files.

---

### Official Review · Reviewer_uRTc · 2022-07-26
**A new dataset for visual correspondence in 2D animation**

**Rating:** 7
**Confidence:** 3
**Correctness:** Yes.

**Strengths:**

* This paper contributes a new dataset specific for visual correspondence tasks in 2D animation. Besides the open dataset, the authors also perform experiments using the new dataset for a benchmark purpose. This motivation and contribution are very fit for this track.

* The dataset contains much more complicated details than the previous dataset for this task. This can be checked in the video, which is a nice add-on.

* Both the pixel-wise and region-wise labels are provided. The experiments also show the effectiveness of both tasks.

* Sufficient details are provided in the main text, there are some minor concerns that I will discuss below.

**Weaknesses:**


* In the experiment, which test set is used to evaluate metrics in Table. 2 and Table. 3? It may be unfair for these comparisons if only evaluated on the test set of AnimeRun, because the boost in the results may come from the less domain gap between train/test data, other than the quality and diversity of the training dataset for better generalization.

Some minor concerns:

* L140: What does it mean to turn off all illuminating objects? Just remove these objects or set the intensity of light to zero? Why may the illuminating objects (e.g., lamps) disappear in the optical flow?

* Fig.4: Why is there no curve of MPI-Sintel in (a) and ATD-12K in (c)?

* L110: "randomly assign representative colors". This part may need more details. For example, how to select representative colors, and how many colors are selected? Any comparison on the statistics before/after this augmentation?

* Why does the FlyingThings3D dataset need to be mixed with the proposed dataset for fine-tuning?

* The optical flow in the video (e.g., about 3:36) changes severely, especially in the background, which seems there are not many changes in the color frame.

* It may be better to perform experiments and provide the evidence for improving the real-world application like frame interpolation to show the potential power of this dataset more clearly.


**Additional Feedback:**

Detail:
* Typo in L258. (3) -> (2)
* Why is the EPE value shown in Fig.6 much larger than the metric in Table.2 ?

**Clarity:**

Overall, the paper is well structured. But the readability still needs some improvement.

**Documentation:**

The dataset is opened, and the code with python API hosted on GitHub seems not ready yet. As shown in the submission instruction, the material like "Hosting, licensing, and maintenance plan", author statement of copyright, and documentation like "dataset of datasheet" is missing.

**Ethics:**

N.A. As the author stats, the proposed dataset is lovely cartoon.

**Relation To Prior Work:**

Yes.

**Summary And Contributions:**

This paper presents a new dataset for visual correspondence of 2D animation, which is generated from three open-source 3D movies. The advantages of utilizing open 3D movies make the components in the film (e.g., background compared to CreativeFlow+[18]) more complicated than the previous dataset. The dataset includes pixel- and region-wise correspondence labels to support some applications like frame interpolation and automatic colorization.

---

> ### Author Response · Authors · 2022-08-18
> **Response to Reviewer uRTc**
>
>
> **Q1: “In the experiment, which test set is used to evaluate metrics in Table. 2 and Table. 3? It may be unfair for these comparisons if only evaluated on the test set of AnimeRun”**
>
> **A1**: The results in Tables 2 and 3 are reported based on the test set of AnimeRun.
>
> Both AnimeRun and CreativeFlow+ are correspondence datasets rendered in 2D cartoon style using Blender. Therefore, they have no domain gap. The differences in performance are mainly due to the quality of the training data. Specifically, the simple image composition (less subjects/objects and less color segments) and small motions in CreativeFlow+ lowers its performance.
>
> Besides, we added an experiment in the Discussion part of the revised paper, where the network finetuned on AnimeRun achieves higher performance on real animation than that on CreativeFlow+.
>
>
> **Q2: “What does it mean to turn off all illuminating objects? Just remove these objects or set the intensity of light to zero? Why may the illuminating objects (e.g., lamps) disappear in the optical flow? ”**
>
> **A2**: It means to set intensity of light to zero and remove emission nodes in material trees. When rendering in 2D style (FLAT lighting under Workbench engine), a material is rendered to its predefined color instead of computing light paths. Therefore, this operation does not affect the rendering of cartoon frames.
>
>
> **Q3: “Fig.4: Why is there no curve of MPI-Sintel in (a) and ATD-12K in (c)?”**
>
> **A3**: Since ATD-12K is a real cartoon dataset that has no ground truth optical flow labels, we are not able to count its flow magnitude. We missed MPI-Sintel in (a) and have added it in the revised paper.
>
> **Q4: “L110: "randomly assign representative colors" needs more details”**
>
> **A4**:  To explain this step, we added an additional part in the revised supplementary file to illustrate the process of color augmentation (Sec. 1).
>
> **Q5: “Why does the FlyingThings3D dataset need to be mixed with the proposed dataset for fine-tuning? ”**
>
> **A5**:  In optical flow experiments, adopting FlyingTings3D is a common practice in network training [1, 2, 3], which can improve performance and generalization. Although it is possible to train on AnimeRun alone, we still apply this practice for its benefits. In the segment matching experiments (Sec. 4.2 in the main paper), AnimeRun is used alone.
>
> **Q6: “The optical flow in the video (e.g., about 3:36) changes severely, especially in the background, which seems there are not many changes in the color frame.”**
>
> **A6**: That is because this kind of visualization is applied to normalized flow, i.e, flow divided by its maximum magnitude [4]. In this clip, the whole scene is shaking irregularly. Although the shaking magnitude is not large, it appears in very different colors, which represent different directions, after normalization.
>
> **Q7: “… provide the evidence for improving the real-world application like frame interpolation …”**
>
> **A7**: We added an experiment in the Discussion part of the revised paper, where we conducted frame interpolation on ATD-12K using optical flow networks finetuned on different datasets to explore the effectiveness of each dataset on real animation data. The experimental results show that AnimeRun improves the quality of interpolated results as a training source.
>
>
> **Q8: “Code is not released yet.”**
>
> **A8**: We have uploaded the code to https://github.com/lisiyao21/AnimeRun.
>
> **Q9: “Documentation is missed.”**
>
> **A9**: We added a datasheet for AnimeRun following the official instructions of NeurIPS dataset track, which is included in the revised supplementary file.
>
>
> [1] Sun et al., Models Matter, So Does Training, TPAMI 2019
>
> [2] Zhou et al., MaskFlowNet: Asymmetric Feature Matching with Learnable Occlusion Mask, ECCV 18
>
> [3] Teed & Deng, RAFT: Recurrent All-Pairs Field Transforms for Optical Flow, ECCV 2020
>
> [4] Baker et al., A Database and Evaluation Methodology for Optical Flow, ICCV 2007

---

> > ### Comment · Reviewer_uRTc · 2022-08-19
> > **Response to authors**
> >
> > Thanks for your response.
> > I think most of my concerns are addressed, thus I decide to raise my rating to reflect this. Specifically, the ATD-12K dataset is used as the test set in the new experiment, which helps to show the power of AnimeRun both on the metrics compared to other competitors and its real-world applicability. Some more details are added to the supplement, and the datasheet is provided.
> >
> > One minor point is not responded yet. Why is the EPE value shown in Fig.6 much larger than the metric in Table.2 ?

---

> > > ### Author Response · Authors · 2022-08-26
> > > **More response to Reviewer uRTc**
> > >
> > > We thank the reviewer for recognizing our work!
> > >
> > > **Q: “One minor point is not responded yet. Why is the EPE value shown in Fig.6 much larger than the metric in Table.2 ?”**
> > >
> > > **A**:
> > > Sorry we missed this question. The example shown in Fig.6 is a hard case with an average magnitude of 103.8 pixels, which is about five times that of the whole dataset;  most parts (99.99%) in this case belong to the “s>50” category in Table.2. Therefore, the EPE values in Fig.6 are much larger than those in the table. We use this example to demonstrate the effectiveness of various methods under large motions and texture-less conditions.

---

### Official Review · Reviewer_WKh3 · 2022-07-27

**Rating:** 8
**Confidence:** 3

**Strengths:**

**[Good Story]**
Authors have a nice story to say. They clearly explain their motivation that has to do with how existing 2D datasets are made and what properties do they lack of. They list the drawbacks of them and come to fix them with their dataset. They argue about their design and label synthesis choices, which completely make sense. They provide a fair and insightful statistical comparison with existing datasets. Finally, they establish a benchmark, draw some concrete conclusions and present a great analysis. While there might be some weaknesses if you zoom in, the big pic is that there is a story well-said.

**[Contribution]**
Authors contribute two-fold.
1. They observe a gap on existing datasets and find an elegant solution of filling it. Their idea is converting 3D movies into 2D animation, so that their dataset has both rich image composition and more complex motion. This can be clearly seen in the video attached.
2. They establish a new benchmark on pixel-wise and region-wise correspondence and provide an insightful analysis.

**[Form]**
The paper is well-written and really easy to follow. Figures are explanatory, video helps presenting the idea and in general one will not find it difficult to follow the paper.

**Weaknesses:**

**[Benchmark]**
The benchmark could have been more rich. Authors choose to evaluate two models for pixel-wise correspondence and one model for region-wise one. While one can draw conclusions from the analysis based on those experiments, it would be probably better to include some more to make it more concrete.

**[Dataset size]**
I guess that the dataset's size is kinda small, am I right? Can it be used alone for training? I am asking because I observed that you also use FlyingThings3D.

**[Real-world applications]**
While you do mention real-world applications and how this dataset can be useful (L13-17), I would suggest explaining it more.

**[Code]**
Code is not released yet, I guess it would be released soon, right?

**Additional Feedback:**

Please consider releasing the code and improving the documentation.

**Clarity:**

The paper is well-written and easy to follow. There is a clear story that unfolds smoothly.

**Correctness:**

In order to verify correctness, it would be crucial to release the code. As it is now, claims and evaluation seem correct. As I have already mentioned, the experiment design is not as rich as it could be in order to make the analysis and conclusion more concrete and to give more value to this dataset.

**Documentation:**

The documentation is poor. Code has not released yet, there is no information about hosting, license and maintenance plan. Please consider adding such info.

**Ethics:**

While the dataset is created using existing 3D movies and probably won't have any ethical implications, it would be nice if authors could clarify this by mentioning it somewhere (e.g. in supplementary). I am mentioning this because there is a slight chance that one of the movies has kind of offensive content (probably not as are lovely cartoons, but just in case...).

**Relation To Prior Work:**

This is one of the paper's strong points. The relation to prior work is exhaustively discussed. The motivation for this dataset is explained.

**Summary And Contributions:**

This paper presents a new 2D animation dataset for pixel-wise (optical flow) and region-wise (segment matching) correspondence. This dataset is made by converting three high-quality 3D movies into 2D animation, thus it is characterized by rich image composition and complex motion. Authors clearly explain he motivation behind this dataset and what makes it different from existing ones. They argue about their design choices, they establish a benchmark and evaluating existing methods on this. Finally, they offer a great analysis that might be useful for future works.

---

> ### Author Response · Authors · 2022-08-18
> **Response to Reviewer WKh3**
>
>
>
> **Q1: “The benchmark could have been more rich”**
>
> **A1**: We conducted additional experiments on two newest optical flow algorithms: GMA [1] (ICCV 2021) and GMFlow [2] (CVPR 2022), and their scores are provided in Table 2. The new experimental results are generally consistent with what we observed in the last version. We also provided additional results in the Discussion section to further verify the merit of AnimeRun.
>
> **Q2: “the dataset's size is kinda small … I observed that you also use FlyingThings3D”**
>
> **A2**: In optical flow experiments, adopting FlyingTings3D is a common practice in network training [3, 4, 5], which can improve performance and generalization. Although it is possible to train on AnimeRun alone, we still apply this practice for its benefits. In the segment matching experiments (Sec. 4.2 in the main paper), AnimeRun is used alone.
>
> The dataset is not “small” considering that its various statistics (summarized in Table 1 of the main paper), e.g., number of segments per frame and occluded segments per frame, are comparatively richer than existing datasets such as CreativeFlow+.
>
>
>
>
> **Q3: “I would suggest explaining it more on real-world applications and how this dataset can be useful (L13-17)”**
>
> **A3**: We added an experiment in the  Discussion section of the revised version, where we conducted frame interpolation on ATD-12K using optical flow networks finetuned on different datasets to explore the effectiveness of each dataset on real animation data. The experimental results show that AnimeRun improves the quality of interpolated results as a training source.
>
>
> **Q4: “Code is not released yet.”**
>
> **A4**: We have uploaded the code to https://github.com/lisiyao21/AnimeRun.
>
>
> **Q5: “The documentation is poor.”**
>
> **A5**: We added a datasheet for AnimeRun following the official instructions of NeurIPS dataset track, which is included in the revised supplementary file.
>
> **Q6: “...there is a slight chance that one of the movies has (some) kind of offensive content…”**
>
> **A6**: All source films of AnimeRun are public content [6,7,8] with no age restriction. While making the data, we have further checked and removed all potentially offensive/fear-inducing content (e.g., snail crawling to ears). We illustrate these issues in the datasheet of revised supplementary files (L.71-77). The provided data should have no such concerns.
>
> [1] Jiang et al., Learning to Estimate Hidden Motions with Global Motion Aggregation, ICCV 2021
>
> [2] Xu et al., GMFlow: Learning Optical Flow via Global Matching, CVPR 2022
>
> [3] Sun et al., Models Matter, So Does Training, TPAMI 2019
>
> [4] Zhou et al., MaskFlowNet: Asymmetric Feature Matching with Learnable Occlusion Mmask, ECCV 2018
>
> [5] Teed & Deng, RAFT: Recurrent All-Pairs Field Transforms for Optical Flow, ECCV 2020
>
> [6] https://www.youtube.com/watch?v=mN0zPOpADL4
>
> [7] https://www.youtube.com/watch?v=SkVqJ1SGeL0
>
> [8] https://www.youtube.com/watch?v=_cMxraX_5RE

---

> > ### Comment · Reviewer_WKh3 · 2022-08-29
> > **Response to authors**
> >
> > Sorry for my late reply. I would like to thank the reviewers for addressing my concerns. You did a good job during the rebuttals and the paper looks really better now, thus I've decided to raise my rating.

---

> > > ### Author Response · Authors · 2022-08-29
> > > **Response to Reviewer WKh3**
> > >
> > > Thank you for recognizing our work!

---

### Official Review · Reviewer_HfEh · 2022-07-27
**Great dataset quality but lack of verification**

**Rating:** 6
**Confidence:** 4
**Correctness:** Seems there is no problem regarding c…
**Clarity:** Clear.

**Strengths:**

- Compared to previous datasets, AnimeRun is larger and composed of better cartoon qualities.
- Easy to follow the paper's claims due to clear writing. Their supplementary materials and experiments clearly evidence the effectiveness of the proposed dataset.
- Great verification about dataset quality.

**Weaknesses:**

- Too few benchmarks to check tendencies of recent models. They have only evaluated two models, i.e., PWC-Net and RAFT. It would be better to evaluate more baseline models to show that their data have similar data distributions to the prior datasets.
- Their proposed data generation methods are limited for transparent objects. Moreover, several effects such as motion blur and bokeh are removed so that these kinds of effects cannot be included in the proposed dataset although they are often found on 2D cartoons.
- Data generation steps are still expensive since they rely on 3D animation datasets. According to the Table 2,  the number of images used for training strongly correlates to the performance. Moreover, the evaluated model achieves better performance when training data are finetuned on FlyingThings3D.  Due to the small dataset size, the AnimeRun cannot be used standalone.
- It uses many external methods; however, their details are not fully given. For better reproducibility, the authors should provide more details to make their paper self-contained.
- I agree that the proposed dataset has many advantages compared to prior datasets. However, I cannot find any real-world applications using 2D cartoon datasets. Why is this dataset needed? Authors should elaborate more about applicability on real-world tasks or add experiments about the usage of the proposed dataset.

**Additional Feedback:**

The work is convincing and the data quality are very good. However, there should be more evaluation results of baseline methods. I would like to happily raise the score if there are more evaluation scores.

**Documentation:**

Yes.

**Ethics:**

Yes.

**Relation To Prior Work:**

Clearly discussed; but as mentioned in the weakness, lack of comparisons of various baseline methods.

**Summary And Contributions:**

- 2D animation data are expensive to collect since they have different characteristics with 3D animations. Simply projecting 3D animations to 2D are not sufficient so that the author proposes a novel approach to make 2D cartoons for 3D.
- Taking benefits from 3D-agnostic methods, they acquire various kinds of labels, which cannot be automatically acquired in 2D cartoons.

---

> ### Author Response · Authors · 2022-08-18
> **Response to Reviewer HfEh**
>
>
> **Q1: “Too few benchmarks to check tendencies of recent models”**
>
> **A1**: We conducted additional experiments on two newest optical flow algorithms: GMA [1] (ICCV 21) and GMFlow [2] (CVPR22), and their scores are provided in Table 2. The new experimental results are generally consistent with what we observed in the last version.
>
>
>
> **Q2: “... data generation methods are limited for transparent object … motion blur and bokeh”**
>
> **A2**:
> Indeed, AnimeRun does not include diverse special effects. As discussed in the revised supplementary file (L. 26-33, Fig. 2), AnimeRun is designed to capture a generic style of 2D cartoons without composition and special effects such as bokeh, shading or blurring. Such properties make AnimeRun well-suited for intermediate stages in the anime production line that involves contour lines and colored frames. It is noteworthy that while lacking such  special effects, AnimeRun is still beneficial for finished animation. In the Discussion part of the revised paper, we show that AnimeRun offers more accurate frame interpolation on real cartoon data than baselines.
>
> Translucent material is not compatible with the optical flow label, since one pixel will have more than one shift. Therefore, we do not include transparent objects following MPI-Sintel.
>
> **Q3: “Data generation steps are still expensive… AnimeRun cannot be used standalone.”**
>
> **A3**:
> In AnimeRun, we aim for richer and more complicated scenes for 2D cartoons than existing dataset. Since the original films were initially created for movie art, they contain many issues unsuitable for 2D animation rendering and obtaining accurate ground truth flow. To ensure the data quality, we had to carefully design a sophisticated data generation pipeline. However, the effort is worthwhile because the synthesized data not only comprise rich image composition but also feature more complex motion patterns, which are beyond existing datasets.
>
> In optical flow experiments, adopting FlyingTings3D is a common practice in network training [3, 4, 5], which can improve performance and generalization. Although it is possible to train on AnimeRun alone, we still apply this practice for its benefits. In the segment matching experiments (Sec. 4.2 in the main paper), AnimeRun is used alone.
>
>
>
> **Q4: “Details of many external methods are not fully given.”**
>
> **A4**: We illustrate the implementation details in the revised supplementary file. We also renewed code on github. See https://github.com/lisiyao21/AnimeRun
>
> **Q5: “Authors should elaborate more about applicability on real-world tasks or add experiments about the usage of the proposed dataset”**
>
> **A5**: We added an experiment in the Discussion part of the revised paper, where we conducted frame interpolation on ATD-12K using optical flow networks finetuned on different datasets to explore the effectiveness of each dataset on real animation data. The experimental results show that AnimeRun improves the quality of interpolated results as a training source.
>
> [1] Jiang et al., Learning to Estimate Hidden Motions with Global Motion Aggregation, ICCV 2021
>
> [2] Xu et al., GMFlow: Learning Optical Flow via Global Matching, CVPR 2022
>
> [3] Sun et al., Models Matter, So Does Training, TPAMI 2019
>
> [4] Zhou et al., Maskflownet: Asymmetric Feature Matching with Learnable Occlusion Mask, ECCV 2018
>
> [5] Teed & Deng, RAFT: Recurrent All-Pairs Field Transforms for Optical Flow, ECCV 2020

---

> > ### Comment · Reviewer_HfEh · 2022-08-29
> > **All my concerns are handled.**
> >
> > Sorry for the late reply and thanks for your comments.
> > I agree many of authors' newly posted statements.
> > They have enhanced the points that I've pointed as weakness.
> > I've decided to raise my score.

---

> > > ### Author Response · Authors · 2022-08-29
> > > **Response to Reviewer HfEh**
> > >
> > > We thank the reviewer's recognition on our work!

---

> ### Author Response · Authors · 2022-08-29
> **Message to Reviewer HfEh**
>
> Dear reviewer,
>
> We appreciate your constructive comments and suggestions to improve our work. We have revised our paper according to your comments. Since it is the last day of the rebuttal period, would you please let us know whether we have resolved your concerns?

---

### Review · Ethics_Reviewer_KrFB · 2022-08-22

**Recommendation:** 1

**Ethics Documentation:**

See the review.

**Ethics Review:**

The dataset is based on a post-processing of publicly available 3D CC-licensed movies. The movies/frames were manually filtered by the authors for "offensive content" but without clear definition of offensiveness. The movies' license and sources are not anywhere neither in the paper nor the supplementary material, including the datasheet.

I recommend that the authors provide the following information:
1. Clearly specify their protocol and details for filtering out the offensive content. What exactly was considered offensive? How were the decisions to remove frames made (frames meet certain criteria -> removed?). How the content screening was done (based on description of movies? Did someone watch all the movies from the beginning till the end?) How many and which movies were censored in the final dataset? This can be done in the datasheet/appendix.
2. Mention the license and the source of the movies (Bender Studio) in the main paper/appendix and the datasheet. Currently, the datasheet only specifies the license of the released dataset, and no information about the sources.
3. If possible, include the list of movies, their sources, metadata such as authors or links where such metadata can be found, in the supplementary material and the website.

---

> ### Author Response · Authors · 2022-08-26
> **Response to Ethics Reviewer KrFB**
>
> We thank the reviewer's constructive comments. We have revised the datasheet to contain more detailed information. New/modified parts are colored to magenta. Responses to the reviewer's concerns are listed below.
>
> **Q1: “Clearly specify their protocol and details for filtering out the offensive content. What exactly was considered offensive? How were the decisions to remove frames made (frames meet certain criteria -> removed?). How the content screening was done (based on description of movies? Did someone watch all the movies from the beginning till the end?) How many and which movies were censored in the final dataset? This can be done in the datasheet/appendix”**
>
> **A1**: The original films to make the dataset are originally public and age-free. They are further filtered by the authors. The protocol is to avoid the content that may be
> - racist/discriminatory/prejudicial;
> - extreme violent/terrifying.
>
> During content screening, if an author disputes the content, we would have a discussion and decide whether to remove it. Since three movies are not too long, we screened the data frame by frame. We have added such descriptions into the datasheet.
>
>
> **Q2: Mention the license and the source of the movies (Bender Studio) in the main paper/appendix and the datasheet. Currently, the datasheet only specifies the license of the released dataset, and no information about the sources.**
>
> **A2**: The licenses of the *Agent327*, *Caminandes*, and *Sprite Fright* are CC-BY 2.0, 3.0, and 1.0, respectively. We have added this to the datasheet.
>
> **Q3: If possible, include the list of movies, their sources, metadata such as authors or links where such metadata can be found, in the supplementary material and the website.**
>
> **A3**: The movies and their metadata links are listed in the main paper (L.46). We have added this into the datasheet and will add them into the website.

---

### Author Response · Authors · 2022-08-18
**General Response to All Reviewers**

We thank all reviewers for their constructive comments and suggestions to our paper. We have revised both the main paper and the supplementary files. New or modified parts are colored to magenta. We also uploaded codes to github (https://github.com/lisiyao21/AnimeRun) and added documentation in supplementary files.

Response to each reviewer has been posted under the reviews. We hope the responses can address reviewers' concerns.

---

### Meta-Review · Area_Chair_VAd8 · 2022-09-11

**Recommendation:** Accept
**Confidence:** 4

**Metareview:**

This paper presents a rich dataset aimed at creating supervision for 2D cartoon tasks. The dataset is created by procedurally converting open source 3D computer graphics movies/shorts into "flattened" 2D frames. Care is taken in describing both the motivation and limitations of this approach. In an area with a dearth of good data, this paper's contributions are warmly welcomed. Reviews appreciated these aspects of the paper while highlighting potential weaknesses that appear addressed in the revisions. I recommend accepting this paper to the NeurIPS 2022 Datasets and Benchmarks program.

---

### Decision · Program_Chairs · 2022-09-16

Accept